# JointMotion: Joint Self-Supervision for Joint Motion Prediction

**Royden Wagner**[1,*]    **Ömer Şahin Taş**[2,*]    **Marvin Klemp**[1]    **Carlos Fernandez**[1]

[1]Karlsruhe Institute of Technology    [2]FZI Research Center for Information Technology

**Abstract:** We present JointMotion, a self-supervised pre-training method for joint motion prediction in self-driving vehicles. Our method jointly optimizes a scene-level objective connecting motion and environments, and an instance-level objective to refine learned representations. Scene-level representations are learned via non-contrastive similarity learning of past motion sequences and environment context. At the instance level, we use masked autoencoding to refine multimodal polyline representations. We complement this with an adaptive pre-training decoder that enables JointMotion to generalize across different environment representations, fusion mechanisms, and dataset characteristics. Notably, our method reduces the joint final displacement error of Wayformer, HPTR, and Scene Transformer models by 3%, 8%, and 12%, respectively; and enables transfer learning between the Waymo Open Motion and the Argoverse 2 Motion Forecasting datasets. Code: https://github.com/kit-mrt/future-motion

**Keywords:** Self-supervised learning, representation learning, multimodal pre-training, motion prediction, data-efficient learning

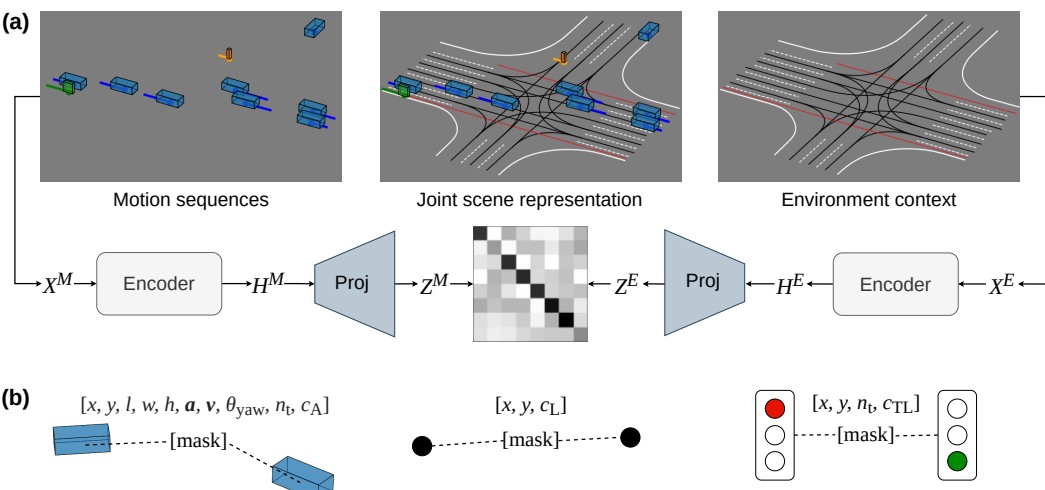

Figure 1: **JointMotion**. **(a)** Connecting motion and environments: Our scene-level objective learns joint scene representations via non-contrastive similarity learning of motion sequences $M$ and environment context $E$. **(b)** Masked polyline modeling: Our instance-level objective refines learned representations via masked autoencoding of multimodal polyline embeddings (i.e., motion, lane, and traffic light data).

---

*Joint first authors. Corresponding author: royden.wagner@kit.edu

8th Conference on Robot Learning (CoRL 2024), Munich, Germany.

# 1 Introduction

Self-supervised learning (SSL) [1] by design excels in applications with large amounts of unlabeled data and limited labeled data (e.g., [2, 3, 4]). However, recent SSL methods combined with supervised fine-tuning outperform plain supervised learning with the same amount of data [5, 6] and shorter training times [7]. This makes SSL a versatile choice to improve existing methods in a wide range of applications.

In this work, we focus on improving joint motion prediction for self-driving vehicles. Motion prediction aims to predict the future motion of traffic agents given their past motion and environment context, such as lane data and traffic light states. Therefore, motion prediction is essential for estimating future interaction among traffic agents [8] and subsequently assessing the risk of planned trajectories [9, 10]. The majority of recent motion prediction methods perform marginal prediction, where future motion is predicted for each agent individually (e.g., [11, 12]). In contrast, joint motion prediction predicts scene-wide motion modes (i.e., capturing a joint set of future motion sequences for multiple agents), which enables interaction modeling.

Since joint motion prediction requires scene-wide representations, we propose a scene-level SSL objective connecting motion and environment. We combine this with an instance-level SSL objective that refines learned representations, enhancing overall prediction accuracy. As scene-level objective, we match past motion sequences in a scene to the corresponding environment context, such as map data and traffic light states. Thereby, a model learns which sets of motion sequences are likely in a given environment, including interaction among agents. As instance-level objective, we reconstruct masked polylines representing past motion sequences, lanes, and past traffic light states. We complement this with an adaptive pre-training decoder, which leads to the high generalizability of our method.

Overall, our main contributions are the design choices that enable our method to generalize across

(a) environment representations (i.e., scene-centric, pairwise relative, and agent-centric) enabled by our complementary scene- and instance-level pre-training objectives,

(b) information fusion mechanisms (i.e., early and late fusion) enabled by the combination of our adaptive pre-training decoder and our instance-level objective, and

(c) dataset characteristics (i.e., varying sequence lengths and feature sets) enabled by the transferability of learned representations.

# 2 Related work

**SSL for motion prediction in self-driving vehicles.** Inspired by the success of SSL in computer vision (e.g., [3, 4, 6]) and natural language processing (e.g., [2, 13, 14]), related methods apply SSL to pre-train motion prediction models for self-driving. PreTraM [15] learns a joint embedding space for trajectory and map data via contrastive learning. During pre-training, the similarity of trajectory and map embeddings from the same traffic scene is maximized, while the similarity to embeddings from other scenes is minimized (i.e., negative examples). This objective is counterproductive when similar scenes are sampled in a mini-batch, e.g., the map embeddings of two four-way stops should be similar, but are erroneously used as negative examples against each other. The concurrent works Traj-MAE [16], Forecast-MAE [17], and RMP [18] are masked autoencoding methods for pre-training motion prediction models. They mask trajectory and/or lane tokens and learn to reconstruct them. Traj-MAE and Forecast-MAE represent motion sequences as trajectory polylines, where each polyline point represents only the agents' position omitting available features such as velocity and acceleration profiles, agent dimensions, yaw angles, and temporal order (see Fig. 1 (b)). Furthermore, they do not include traffic light states as environment context. SSL-Lanes [19] extends lane masking with the objectives of distance to intersection prediction, maneuver and success/failure classification. Although these objectives are well adapted to motion prediction, they require non-trivial heuristics (e.g., for clustering maneuvers).

**Joint motion prediction for self-driving vehicles.** Joint motion prediction aims to predict the joint distribution of future motion sequences over multiple agents in a traffic scene. Thus, a predicted motion mode represents a scene-wide set of motion sequences with one sequence per agent. Scene Transformer [20] is an encoder-decoder transformer model for joint motion prediction. The encoder learns global scene-centric representations (i.e., agent, lane, and traffic light features), the decoder transforms these embeddings into joint motion modes. Global position embeddings are used for the learned scene-centric representations, which are more difficult to learn than pairwise relative position embeddings (cf. [21, 22]). MotionLM [23] reformulates motion prediction as language prediction task and learns a vocabulary of discrete motion vectors. Joint prediction modes are generated by autoregressively decoding sequences of motion vectors for multiple agents. MotionDiffuser [24] performs joint motion prediction as conditional denoising diffusion process. A set of noisy trajectories (i.e., positions disturbed by Gaussian noise) is transformed by a denoiser into a set of trajectories that approximates the ground truth joint prediction mode. The learned denoiser is conditioned on environment context such as lane data and traffic light states. Both MotionLM and MotionDiffuser use an agent-centric encoder [12], which leads to repeated computation since a traffic scene is encoded for each agent individually. Furthermore, their best performing variants exhibit high inference latency (MotionLM: $250\,\text{ms}^2$, MotionDiffuser: $409\,\text{ms}$).

**Marginal motion prediction with auxiliary interaction prediction objectives.** A marginal motion mode represents a single motion sequence for one agent. Many recent methods extend marginal prediction models with modules for interaction prediction objectives (i.e., dense [25, 26, 27] or conditional prediction [28, 29]). In such methods, the main prediction modules are still trained in a marginal manner, where training targets are marginal motion modes per agent. The additional modules are trained to combine marginal predictions or the corresponding latent representations by minimizing overlap between them. Thus, these methods do not learn to model scene-wide joint motion modes from ground up, but how to combine marginal modes with less overlap. This limits interaction modeling to the learned re-combination of marginal motion predictions.

## 3 Method

We present our method for self-supervised pre-training of motion prediction models in two steps. First, we describe our scene-level SSL objective (connecting motion and environments), then, our instance-level SSL objective (masked polyline modeling) and the adaptive pre-training decoder.

### 3.1 Connecting motion and environments

Joint motion prediction requires scene-wide representations to decode joint modes, which include predictions of multiple agents within a scene. Therefore, we propose pre-training motion prediction models by connecting motion and environments (CME). This scene-level objective aims to learn a joint embedding space for motion sequences and environment context (see Figure 1 (a)). Thus, a model implicitly learns which motion is likely in a given environment, including traffic rules and interaction among agents.

In detail, we use the past motion of all agents within a scene to generate a scene-level motion embedding ($Z^M$ in Figure 1) and combine lane data and traffic light states to a corresponding environment embedding ($Z^E$ in Figure 1). These embeddings are generated by modality-specific encoders (i.e., for motion, lanes, and traffic lights) followed by global average pooling and an MLP-based projector (Proj in Figure 1). We perform average pooling on the intermediate embeddings ($H^M$ and $H^E$ in Figure 1) to be invariant to variations in the number of agents and the complexity of environments. Following [30], we use two separate MLPs with LayerNorm and ReLU activations as projectors, each having a hidden dimension of 2048 and an output dimension of 256. We use the modality-specific encoders of recent motion prediction models (e.g., [20, 21]) without modifications and remove the additional projector after pre-training. The joint embedding space is learned

---

[2]This latency is linearly extrapolated, as inference latency is only reported up to the second best/largest model (see Table 8 in the Appendix of [23])

by similarity learning via redundancy reduction. Following [31, 32], we reduce the redundancy of vector elements per embedding (i.e., for $Z^M$ and $Z^E$ individually) and maximize their similarity by approximating the cross-correlation matrix $C$ of $Z^M$ and $Z^E$ to the corresponding identity matrix:

$$\mathcal{L}_{\text{CME}} = \lambda_{\text{red}} \sum_i \sum_{j \neq i} C_{ij}{}^2 + \sum_i (1 - C_{ii})^2,$$

where $i, j$ index the vector dimension of the embeddings $Z$. The redundancy reduction term is scaled by $\lambda_{\text{red}}$ and ensures that individual embedding elements capture different features. Therefore, it prevents representation collapse with trivial solutions for all embeddings across all scenes (e.g., zero vectors). Non-trivial yet identical solutions are not explicitly prevented, but are unlikely, as empirically shown in [31]. In contrast to [15], our scene-level objective does not require negative examples, which are difficult to define in this context. Unlike [32], this objective maximizes the similarity of embeddings from different modalities (i.e., motion and environment) rather than augmented views of the same modality, removing the requirement to develop suitable augmentations.

## 3.2 Masked polyline modeling with adaptive decoding

Scene-level representations are well suited to provide an overview of traffic scenes, but lack instance-level details. For example, the exact position of traffic agents in the past or lane curvatures. Therefore, we combine our scene-level objective with the instance-level objective of masked polyline modeling (MPM) to refine learned representations.

Inspired by masked sequence modeling [2, 6], we mask elements of polylines representing past motion sequences, lanes, and past traffic light states and learn to reconstruct them from non-masked elements and environment context. As shown in Figure 1, we represent traffic agents with 10 features rather than just past positions (cf. [16, 17]). In detail, we reconstruct agent positions $(x, y)$, dimensions $(l, w, h)$, acceleration $\boldsymbol{a}$, velocity $\boldsymbol{v}$, yaw angle $\theta_{\text{yaw}}$, temporal order $n_{\text{t}}$, and classes $c_{\text{A}}$ (i.e., vehicles, cyclists, and pedestrians). For lanes, we reconstruct positions $(x, y)$ and lane classes $c_{\text{L}}$. For traffic light state sequences, we reconstruct positions $(x, y)$, temporal order $n_{\text{t}}$, and state classes $c_{\text{TL}}$ (i.e., green, yellow, and red). Following [21], positions, dimensions, accelerations, velocities, and yaw angles are represented as float values and temporal order and agent classes as boolean one-hot encodings.

For models with late or hierarchical fusion mechanisms (e.g., [20, 21]), we use the modality-specific encoders to generate embeddings per modality (see Figure 2 (a)). Afterwards, we concatenate these embeddings and use a shared local decoder to reconstruct masked sequence elements from non-masked elements and context from other modalities. As local decoder, we use transformer blocks with PreNorm [33], local attention [34] with an attention window of 32 tokens, 8 attention heads, rotary positional embeddings [35], and FeedForward layers with an input and hidden dimension of 256 and 1024.

For models that employ early fusion mechanisms (e.g., [12]), we use learned queries and a shared decoder to reconstruct the input sequences (see Figure 2 (b)). Such models learn a compressed latent representation for multi-modal input. Therefore, we use learned queries in the same number as input tokens to decompress these representations and reconstruct the input sequences. We use a regular cross-attention mechanism between the learned queries and compressed latent representations and a local self-attention mechanism within the set of learned queries. The resulting transformer blocks have the same hyperparameters as in the late fusion setup.

For both variants, we use random attention masks for masking and a masking ratio 60%. As training target, we minimize the Huber loss between the reconstructed polylines and the input polylines $\mathcal{L}_{\text{MPM}} = \lambda_{\text{A}} \mathcal{L}_{\text{A}} + \lambda_{\text{L}} \mathcal{L}_{\text{L}} + \lambda_{\text{TL}} \mathcal{L}_{\text{TL}}$. If not specified otherwise, we set $\lambda_{\text{A}} = \lambda_{\text{L}} = \lambda_{\text{TL}} = 1$.

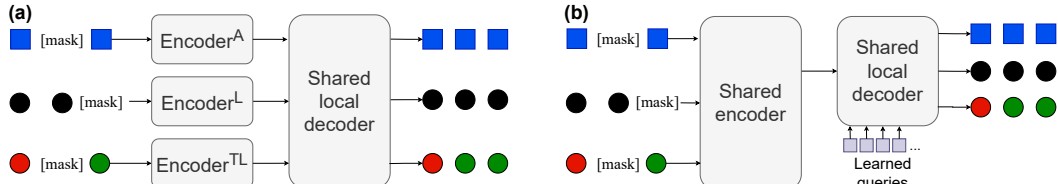

Figure 2: **Adaptive decoding for masked polyline modeling with late and early fusion encoders.** (a) Late fusion with modality-specific encoders for agents (Encoder$^A$), lanes (Encoder$^L$), and traffic lights (Encoder$^{TL}$). (b) Early fusion with a shared encoder for all modalities. Compressed features are decoded using learned query tokens.

## 4   Experiments

### 4.1   Comparison of self-supervised pre-training methods for joint motion prediction

In this experiment, we compare our method with recent self-supervised pre-training methods for autonomous driving using the WOMD dataset. Specifically, we compare with contrastive learning via PreTraM [15] and masked autoencoding with Forecast-MAE [17] and Traj-MAE [16].

**Motion prediction models.** We pre-train and fine-tune well-established Scene Transformer [20] models on the WOMD training split. We use the publicly available implementation by [21] with 3 modality-specific encoders (i.e., for agents, lanes, and traffic lights). Adapted to the complexity per modality, we encode traffic light features with 1, agent features with 3, and lane features with 6 transformer blocks.

**Pre-training.** For all methods, we add a pre-training decoder for late fusion models (see Figure 2) with 3 transformer blocks and the hyperparameters described in Section 3.2. For our method, we additionally add two projectors for scene-level representations as described in Section 3.1. For Pre-TraM, we follow its trajectory-map contrastive learning configuration and add two linear projection layers as projectors for trajectory and map embeddings. For Forecast-MAE, we use a masking ratio of 60% and reconstruct positions of lane polylines and past trajectories by minimizing the MSE loss. We exclude future trajectories, since self-supervised learning by design does not use the same labels as the intended downstream task (cf. [1]). For Traj-MAE, we use a masking ratio of 60% and reconstruct positions of lane polylines and past trajectories by minimizing the corresponding Huber loss. For our method, we minimize the joint loss of our proposed objectives $\mathcal{L}_{\text{JointMotion}} = \lambda_{\text{CME}}\mathcal{L}_{\text{CME}} + \mathcal{L}_{\text{MPM}}$. We set $\lambda_{\text{CME}} = 0.01$ and following [31] the weight of the redundancy reduction term $\lambda_{\text{red}} = 0.005$.

**Fine-tuning.** For all methods, we replace the pre-training decoder with a shared global decoder and learned anchors for $k = 6$ motion modes. We initialize the modality-specific encoders with the learned weights from pre-training and do not freeze any weights during fine-tuning. We fine-tune the model using its joint configuration and hard loss assignment. Accordingly, the loss is computed for the best scene-wide joint prediction mode. As post-processing, we follow [36] and adjust the confidences of redundant predictions.

**Training time, hardware, and optimizer.** For all methods, we perform pre-training for 10 hours and fine-tuning for 23.5 hours using a training server with 4 A100 GPUs. For pre-training and fine-tuning, we use AdamW [37] with an initial learning rate of 1e-4 and a step learning rate scheduler with a reduction rate of 0.5 and a step size of 25 epochs.

**Results.** Table 1 shows the results of this experiments. Explicit scene-level objectives (i.e., Pre-TraM and JointMotion) lead to better and more balanced performance across all agent types, while implicit scene-wide masked autoencoding with Forecast-MAE or Traj-MAE tends to focus more on the pedestrian class than on the others (see mAP scores). Therefore, it is likely that explict scene-wide objectives enforce learning interactions between varying agent types more. Traj-MAE and our method without the objective of connecting motion and environments (JointMotion w/o CME)

achieve better scores than Forecast-MAE. Consequently, reconstructing individual elements of poly-lines improves representations more than reconstructing whole polylines. The superior performance of JointMotion w/o CME compared to Traj-MAE indicates that extending the motion feature set and reconstructing traffic light states as well further improves learned representations. Our method without masked polyline modeling (JointMotion w/o MPM) performs on par with PreTraM, indicating that redundancy reduction can replace negative examples for scene-level similarity learning.

| Pre-training | mAP↑ | | | | minADE↓ | | | minFDE↓ | | |
|---|---|---|---|---|---|---|---|---|---|---|
| | avg | cyc | ped | veh | cyc | ped | veh | cyc | ped | veh |
| None | 0.1596 | 0.1465 | 0.1821 | 0.1504 | 1.522 | 0.698 | 1.486 | 3.653 | 1.654 | 3.476 |
| Forecast-MAE [17] | 0.1592 | 0.1420 | 0.1873 | 0.1482 | 1.529 | 0.694 | 1.423 | 3.664 | 1.652 | 3.343 |
| Traj-MAE [16] | 0.1677 | 0.1492 | 0.1917 | 0.1623 | 1.421 | 0.708 | 1.338 | 3.320 | 1.647 | 3.090 |
| PreTraM [15] | 0.1689 | 0.1724 | 0.1775 | 0.1569 | 1.488 | 0.711 | 1.461 | 3.489 | 1.657 | 3.374 |
| JointMotion w/o MPM | 0.1689 | 0.1762 | 0.1777 | 0.1528 | 1.478 | 0.684 | 1.406 | 3.507 | 1.608 | 3.287 |
| JointMotion w/o CME | 0.1784 | 0.1652 | 0.1903 | 0.1796 | 1.457 | 0.700 | 1.317 | 3.363 | 1.630 | 3.033 |
| JointMotion | **0.1940** | **0.1970** | **0.1964** | **0.1886** | **1.343** | **0.677** | **1.288** | **3.095** | **1.583** | **2.941** |

Table 1: **Comparison of self-supervised pre-training methods for joint motion prediction.** All methods are used to pre-train Scene Transformer models [20] on the Waymo Open Motion dataset and evaluated on the validation split. Agent types: cyclist (cyc), pedestrian (ped), and vehicle (veh). Best scores are **bold**, second best are underlined.

Overall, pre-training with both proposed objectives (i.e., JointMotion) leads to the best scores across all agent types. This shows that the combination of our objectives works best. Figure 3 further highlights the complementary nature of our two objectives. Specifically, the reconstruction of past traffic light sequences are learned very similar with both configurations. The reconstruction loss for past agent motion converges more slowly with JointMotion pre-training, but reaches similar values as well. However, with our scene-level objective, the lane reconstruction loss likely converges to a higher value. We hypothesize that models pre-trained with our scene-level objective tend to focus more on the overall lane structure than on specific details of individual lane polylines.

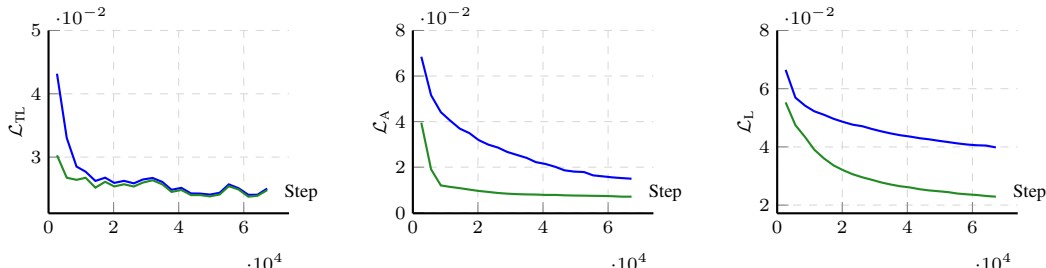

Figure 3: **Loss plots of our complementary pre-training objectives**. The green curve represents JointMotion w/o CME, while the blue curve represents JointMotion. Consistent with the remainder of the document, L stands for lanes, TL stands for traffic lights, and A stands for agents.

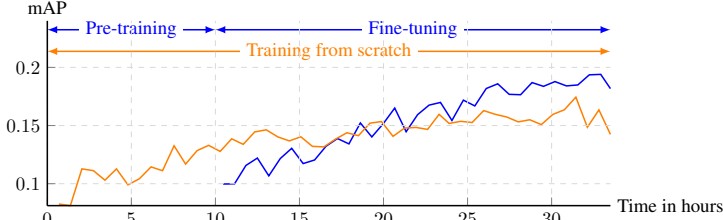

Figure 4: **Accelerating and improving training via SSL.** Scene Transformer models pre-trained with JointMotion achieve higher mAP scores on WOMD than models trained from scratch.

Figure 4 shows that Scene Transformer models pre-trained with our method achieve higher mAP scores on WOMD than models trained from scratch, even in a shorter wall training time.

## 4.2 Comparing scene-level self-supervision methods

In this experiment, we further compare the scene-level objectives PreTraM [15] and JointMotion. We train Scene Transformer, HPTR, and a joint configuration of Wayformer to cover all common types of environment representations in motion prediction (i.e., scene-centric, pairwise relative, and agent-centric).

**Experimental setup.** For Scene Transformer, we use the same configuration as in the previous experiment. For HPTR, we analogously add 3 modality-specific encoders and use a shared decoder for $k = 6$ motion modes. For the joint configuration of Wayformer, we follow [24] and use a shared encoder for early fusion, which compresses the multi-modal input to 128 tokens. We concatenate multiple such agent-centric embeddings with positional and rotation information into a common reference frame and use a shared decoder to predict joint motion modes. For pre-training the Wayformer model, we add our decoder for early fusion configurations (see Figure 2). The Wayformer model is not pre-trainable with PreTraM since an instance-level objective is required to decode the modality-specific tokens from fused representations (cf. Section 3.2). For all models, we employ the same hardware, training time, optimizer, and learning rate scheduling as in the previous experiment. Evaluation is performed on the interactive validation split of WOMD and AV2.

**Results.** Table 2 shows the results of this experiment. Our method consistently outperforms Pre-TraM using different models with varying environment representations and fusion mechanisms. Our method improves all models, while the improvement of the scene-centric Scene Transformer model (e.g., 12% lower minFDE) is most significant and the improvement of the agent-centric Wayformer model is least significant (e.g., 3% lower minFDE). Hence, the improvements are inversely proportional to the sample efficiency of the models. In scene-centric modeling, one sample is generated per scene, whereas in agent-centric modeling, one sample is generated for each traffic agent within a scene. This aligns with the finding that fine-tuning with more samples generally reduces the value of pre-training [38].

| Dataset | Model (config) | Pre-training | minFDE↓ | | minADE↓ | | MR↓ | | OR↓ | |
|---|---|---|---|---|---|---|---|---|---|---|
| WOMD | Scene Transformer | None | 3.6715 | | 1.5255 | | 0.7372 | | 0.2868 | |
| | | PreTraM [15] | 3.6508 | -0.57% | 1.5415 | 1.05% | 0.7385 | 0.18% | 0.2915 | 1.64% |
| | | JointMotion | **3.2400** | -11.75% | **1.3830** | -9.36% | **0.7090** | -3.82% | **0.2847** | -0.73% |
| | HPTR | None | 2.6003 | | 1.1682 | | 0.6030 | | 0.2331 | |
| | | PreTraM [15] | 2.5049 | -3.66% | 1.0981 | -5.99% | 0.5863 | -2.78% | 0.2345 | 0.60% |
| | | JointMotion | **2.4006** | -7.68% | **1.0564** | -9.58% | **0.5591** | -7.28% | **0.2297** | -1.46% |
| | Wayformer (joint) | None | 2.3529 | | 1.0209 | | 0.5461 | | 0.2273 | |
| | | JointMotion | **2.2823** | -3.00% | **0.9939** | -2.64% | **0.5270** | -3.50% | **0.2143** | -5.72% |
| AV2 | HPTR | None | 2.2550 | | 1.1380 | | - | | **0.0988** | |
| | | JointMotion WOMD | **2.1530** | -4.53% | **1.1370** | -0.09% | - | | 0.1025 | 3.75% |

Table 2: **Comparing scene-level self-supervision methods.** All metrics are computed using the Waymo Open Motion interactive (WOMD) and Argoverse 2 Forecasting (AV2) validation splits. Best scores are **bold**.

Unlike PreTraM, the proposed adaptive pre-training decoder combined with our instance-level objective enable our method to adapt to models with early fusion mechanisms (e.g., Wayformer). Specifically, modality-specific masked polyline modeling with learned queries in the same number as input tokens enables our method to decode modality-specific tokens (i.e., agent, lane, and traffic light) from compressed latent representations. This is particular relevant since the current state-of-the-art methods on the Argoverse 1 Forecasting (ProphNet [11] via AiP tokens) and Argoverse 2 Forecasting (QCNeXt [39] via query-centric modeling) benchmarks rely on fusion mechanisms with compressed latent representations as well.

Furthermore, our pre-training leads to comparable improvements on the interactive and the regular validation splits (cf. Table 1), while pre-training with PreTraM leads to smaller improvements on the interactive validation split. We hypothesise that with our additional instance-level objective, more fine-grained trajectory details are learned, which is more important for close trajectories of

interacting agents. The lowest block shows that our method leads to transferable representations. Specifically, pre-training on WOMD improves fine-tuning on AV2.

## 4.3 Comparison with state-of-the-art methods for joint motion prediction

In this experiment, we compare our method with state-of-the-art methods for joint motion prediction in autonomous driving.

**Experimental setup.** We pre-train HPTR (configured as in Section 4.2) for 10 hours using JointMotion and fine-tune for 100 hours. We evaluate the methods in test and validation splits of WOMD. We use the official challenge website to compute performance metrics.

**Results.** Table 3 presents a comparative analysis of various state-of-the-art methods for joint motion prediction on interactive splits of WOMD. In the test split, we compare four different approaches. The Scene Transformer has a mAP of 0.1192 and is outperformed by the other methods in all metrics. GameFormer achieves a higher mAP of 0.1376 and exhibits competitive performance in terms of minADE and minFDE. MotionDiffuser and JointMotion (HPTR) are close contenders, with MotionDiffuser performing slightly better in all of the metrics compared to JointMotion (HPTR).

For the validation split, the results are similar. GameFormer (joint) scores a mAP of 0.1339, with MotionLM (single replica) outperforming it in terms of mAP, achieving 0.1687. JointMotion (HPTR) presents an improved mAP of 0.1761 compared to its counterparts. Following [21], we compare against the single replica model of MotionLM and show its ensembling version for reference. MotionLM (ensemble) demonstrates superior overall performance, as expected given the increased modeling capacity.

| Split | Method (config) | Venue | mAP↑ | minADE↓ | minFDE↓ | MR↓ | OR↓ |
|-------|-----------------|-------|------|---------|---------|------|------|
| Test | Scene Transformer (joint) [20] | ICLR'22 | 0.1192 | 0.9774 | 2.1892 | 0.4942 | 0.2067 |
| | GameFormer (joint) [40] | ICCV'23 | 0.1376 | 0.9161 | **1.9373** | 0.4531 | 0.2112 |
| | MotionDiffuser [24] | CVPR'23 | **0.1952** | **0.8642** | 1.9482 | **0.4300** | **0.2004** |
| | JointMotion (HPTR) | | 0.1869 | 0.9129 | 2.0507 | 0.4763 | 0.2037 |
| Val | GameFormer (joint) [40] | ICCV'23 | 0.1339 | **0.9133** | **1.9251** | **0.4564** | - |
| | MotionLM (single replica) [23] | ICCV'23 | 0.1687 | 1.0345 | 2.3886 | 0.4943 | - |
| | JointMotion (HPTR) | | **0.1761** | 0.9689 | 2.2031 | 0.4915 | **0.1990** |
| | MotionLM (ensemble) | ICCV'23 | 0.2150 | 0.8831 | 1.9825 | 0.4092 | - |

Table 3: **Comparison with state-of-the-art methods for joint motion prediction.** All methods are evaluated on interactive splits of the Waymo Open Motion dataset. Following [21], we compare against single replica versions of joint prediction methods, ensemble versions are shown for reference. Best scores are **bold**, second best are underlined.

## 5 Conclusion

In this work, we introduced a SSL method for joint motion prediction of multiple agents within a traffic scene. Our SSL framework integrates scene-level and instance-level objectives that operate complementarily, enhancing the training speed and accuracy of motion prediction models. Notably, JointMotion outperforms recent contrastive and autoencoding methods for pre-training in motion prediction. Moreover, our method generalizes across different environment representations, information fusion mechanisms, and dataset characteristics. Our evaluations demonstrate significant performance improvements over non-ensembling joint prediction methods or joint-prediction variants of marginal-prediction architectures, underscoring the robustness and effectiveness of our proposed method.

**Limitations.** Our method is not as label-efficient as SSL methods in computer vision, which generate supervisory signals from raw images (e.g., [3]). Our pre-training does not include the downstream labels for motion prediction (i.e., future motion), but relies on past motion and map data.

**Acknowledgments**

The research leading to these results is funded by the German Federal Ministry for Economic Affairs and Climate Action within the project "NXT GEN AI METHODS – Generative Methoden für Perzeption, Prädiktion und Planung". Furthermore, we acknowledge the financial support by the German Federal Ministry of Education and Research (BMBF) within the project HAIBrid (FKZ 01IS21096A).

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

# Appendix

**Evaluation metrics.** Following [41], we use the mean average precision (mAP), the average displacement error (minADE), the final displacement error (minFDE), miss rate (MR), and overlap rate (OR) to evaluate motion predictions. All metrics are computed using the minimum mode for $k = 6$ modes. Accordingly, the metrics for the mode closest to the ground truth are measured. We use the official challenge website to compute metrics on WOMD. For the AV2 dataset, we use the evaluation software provided by [21]. Therefore, joint modes (best scene-wide mode) are evaluated on the interactive splits and marginal modes (best mode for each agent individually) on the regular splits.

**The Waymo Open Motion Dataset (WOMD)** [41] is comprised of over 1.1 million data points extracted from 103,000 urban or suburban driving scenarios, spanning 20 seconds each. The state of object-agents includes attributes like position, dimensions, velocity, acceleration, orientation, and angular velocity. Each data point captures 1 second of past followed by 8 seconds of future data. We resample this time interval with 10Hz.

**The Argoverse 2 Motion Forecasting Dataset (AV2)** [42] is comprised of 250,000 urban or suburban driving scenarios each spanning 11 seconds, with 5 seconds of past and a 6 seconds of future data. The datasets entails interactions in over 2,000 km of roadways across six geographically diverse cities.

