# OpenReview forum: "JointMotion: Joint Self-Supervision for Joint Motion Prediction"
_robot-learning.org/CoRL/2024/Conference — CoRL 2024_

### Official Review · Reviewer_XhvA · 2024-07-01
**Incremental Advancement of Using Self-supervision Learning for Joint Motion Prediction**

**Originality:** 3
**Technical Quality:** 3
**Clarity Of Presentation:** 2
**Potential Impact:** 3
**Recommendation:** 3
**Confidence:** 4

**Review:**

Before the review content I’d like to add some background glossary to disambiguate for those unfamiliar.
- Joint vs Marginal: The “Joint Motion Prediction” terms refers to motion prediction models that output non-conflict, interaction-aware trajectories for all agents in a scene. In contrary, marginal prediction refers to the models that output trajectory for each agent. Such models can still make predictions for each agent in the scene, but their consideration for interactions is limited and might introduce conflicts. With the terminology clarified, we can see the need for a good joint predictor, and part of this is a good scene centric representation, which the author claim can be improved with SSL.
- Scene level vs instance level representation:  In the context of this article, scene level representation encodes the interactions and relationships among all agents and environmental features, whereas instance level encodes detailed attributes and features of individual agents and environmental elements, focusing on fine-grained information.

### Quality
- [Strength] Ample experiments: The author compared against SSL baselines who (1) only used scene-level similarity objective (3) only used masked autoencoding. The author also show JointMotion’s generalizability on the backbone model by experimenting with (1) Scene Transformer, (2) HPTR, and (3) Wayformer, which are industry recognized models.
- [Weakness] The author did not seem to experiment ablating the masked autoencoding from JointMotion.

### Clarity
- [Strength] The article is divided into clear sections, allow readers to follow through.
- [Weakness] The procedure for pre-train deserves more description. For example, when you use JointMotion on Scene Transformer, do you freeze any weights at fine-tune? What data do you use in pre-train vs fine-tune? During pre-train do you train the scene-level and instance-level separately then copy the weights to the backbone model to further fine-tune?
- [Weakness] In section 1 the clarity of your summary of contributions can be improved by drawing a clear relationship between each of your design choice and its improvement, such as saying the scene similarity learning improved balance performance across agent types.
- [Weakness] Section 4.2 Experiment setup needs clarity improvement: correct me if I’m wrong. I infer from the context that you base your JointMotion and the other baselines on top of the Scene Transformer by adding the CME, masked autoencoding and/or decoder. At line 173, you mentioned you used pre-training, is this only applied to JointMotion? By placing the sentence “For pre-training…” before the next sentence “For our method…”, you make the former sentence seem like pre-training is not part of your method but rather applied to the baselines as well.
### Originally
- [Strength] Though both scene level similarity learning and masked autoencoding methods have been applied to motion prediction, this work is the first to combine them and suggests their complementary property. This contributes to the field of autonomous driving.
- [Weakness] While the combination of techniques is novel, the individual components are extensions of existing methods. The originality might be seen as incremental rather than groundbreaking.

### Significance
- [Strength] The reported improvements in trajectory metrics (mAP, minADE etc) are significant as compared to other SSL methods, suggesting the effectiveness of the method.
- [Weakness] The reported improvement as compared to SOTA prediction methods falls short in various metrics. Thus in Table 3: It might be helpful to include the number of parameters in your model and the others to show the difference in model capacity.

**Quality Of The Limitations Section:**

3

**Questions For Rebuttal:**

1. In section 1 contribution c: by saying your method can generalize across datasets, how many different datasets did you experiment? I only see WOMD and Argoverse 2. In addition, saying a model that can work on two datasets is generalizable is not rigor, unless you are talking about zero-shot generalizability, which does not seem the case in your work.
2. Page 2 line 27: you listed reference 11 (Motion Transformer) as an example of marginal prediction model without interaction modeling. However, in section 3.1 of that paper, the author stated their model perform interaction prediction as part of the scene context encoding.
3. Page 5 line 181: typo `ration`->`ratio`.
4. Page 5 line 184, do you intent to include $\lambda_{red}$ in the total loss defined at line 183?
5. Page 5 line 196, there is a typo where you wrote 2x `without`.
6. Page 6 line 205: how do you conclude lane reconstructions are more accurate in this case? Is figure 3 a plot of the training time loss? Indeed the blue curve decreases slower but they do not seem fully converged yet. You can say without scene-level objective, the training loss of lane reconstruction converges slower than that of agent state and traffic light but I don’t think it is evident enough to conclude lane reconstruction is less accurate.
7. You have experiments ablating the CME from JointMotion. Do you also have experiments ablating masked autoencoding and compare with the scene level SSL method PreTraM?
8. Page 6 figure 4: I’m a little confused by this figure. If I understand correctly, the orange curve and blue curves are meant to represent the $mAP$ progress under the same training time? First, your x-axis is step rather than time. Second, the orange curve (train from scratch) takes $1.8e5$ steps whereas the blue curve (pre-train + fine-tune) takes $0.8e5$ steps for pre-train plus $1.8e5$ steps for fine-tuning. Only the fine-tuning part should have taken the same time as the entire orange part given the model is the same, right? Are you sure the train-from-scratch configuration is given the same amount of time as the pre-train-then-fine-tune configuration?

**Robotics Focus:**

3

**Summary Of Paper:**

This paper JointMotion brings another way to employ data-efficient self-supervised learning (SSL) pre-training method to enhance existing autonomous driving joint motion prediction models. Unlike previous works, in this work, SSL is applied in two-fold to: (1) non-contrastive similarity learning for past motion sequences and environmental context at the scene level, creating robust joint scene representations (in section 4 this is also called Connection motion and environment (CME)); and (2) masked autoencoding and adaptive decoding to refining multimodal polyline representations, including motion, lanes, and traffic lights at the instance level. The authors’ experiment show this method outperforms existing SSL methods and is able to generalize across backbone models.

**Summary Of Recommendation:**

The contribution is incremental. The article needs multiple clarifications.

---

### Official Review · Reviewer_ZYgU · 2024-07-11
**Self-supervision learning for joint motion prediction**

**Originality:** 4
**Technical Quality:** 3
**Clarity Of Presentation:** 4
**Potential Impact:** 3
**Recommendation:** 3
**Confidence:** 4

**Review:**

Strengths:
This paper explores self-supervised learning at both the scene level and instance level to combine motion and map information for joint motion prediction.

The paper shows that this self-supervised learning approach can be effectively applied across different environment representations and information fusion mechanisms on both commonly used benchmark datasets, including the Waymo open motion dataset and the Argoverse 2 motion forecasting dataset.

The paper is well-written and the results are quite promising.

Weaknesses:
The methodology lacks some details, which require readers to be familiar with the detailed settings of models [21], [22], [31], and [32]. It would be easier for readers to summarize such details in the supplementary material.

The description of Figure 3 is not accurate. I do not see a strong similarity in the reconstruction of past motion with both configurations. The explanation of the disparity in the lane reconstruction with both configurations is not clear and convincing to me.

In Figure 1, the training from scratch stops even though it still depicts an increasing trend. It is not clear why the training from scratch is not compared with fine-tuning up to the same total number of training steps.

It seems that the experimental settings for Tables 2 and 3 are different since the numbers for JointMotion (HPTR) on WOMD are different. The authors should clarify these differences.

**Quality Of The Limitations Section:**

2

**Questions For Rebuttal:**

Overall, the paper shows the potential of exploring self-supervised learning at both the scene level and instance level to combine motion and map information for joint motion prediction. But the weaknesses mentioned above should be addressed.

**Robotics Focus:**

3

**Summary Of Paper:**

This paper proposes joint self-supervision learning for joint motion prediction. It employs scene-level self-supervised learning to connect motion and environment information from past trajectories of each agent and HD map polylines. The scene-level self-supervised learning is further combined with instance-level self-supervised learning to reconstruct masked trajectories and map polylines to improve prediction accuracy.

**Summary Of Recommendation:**

The paper has some drawbacks but they can be probably well addressed in the rebuttal. Hence, my rating is a weak accept

---

### Official Review · Reviewer_ffHe · 2024-07-15
**Review for submission 284**

**Originality:** 2
**Technical Quality:** 3
**Clarity Of Presentation:** 2
**Potential Impact:** 2
**Recommendation:** 3
**Confidence:** 3

**Review:**

Strengths:
* Scene-level Objective: The paper proposes a scene-level SSL objective that connects motion and environment. By matching past motion sequences to the corresponding environment context, the model learns which sets of motion sequences are likely in a given environment, including the interaction among agents. This objective does not require negative examples and maximizes the similarity of embeddings from different modalities, enhancing the model's ability to learn scene-wide representations.
* Instance-level Objective: The paper combines the scene-level objective with the instance-level objective of masked polyline modeling. By reconstructing masked polylines representing past motion sequences, lanes, and past traffic light states, the model refines the learned representations and captures more details at the instance level.
* Adaptive Pre-training Decoder: The paper introduces an adaptive pre-training decoder that enables the method to generalize across different environment representations, information fusion mechanisms, and datasets. This decoder allows the model to adapt to models with early fusion mechanisms and decode modality-specific tokens from compressed latent representations.
* Experimental Results: The paper conducts extensive experiments to compare JointMotion with recent self-supervised pre-training methods and state-of-the-art methods for joint motion prediction. The results show that JointMotion outperforms other methods and leads to significant performance improvements over non-ensembling joint prediction methods or joint-prediction variants of marginal-prediction architectures.


Weaknesses:

* It appears that utilizing similarity learning to acquire the joint embedding space for motion and environments may not be considered a major contribution as this kind of approach has been extensively studied in [23].
* As listed in the limitation part, future trajectories should also be valuable for pre-training, at least same importance as history information.

**Quality Of The Limitations Section:**

1

**Questions For Rebuttal:**

* Could you kindly elaborate on why explicit scene-level objective methods typically tend to yield better and more balanced performance across all agent types in comparison to implicit methods?
* Could you please explain why there is a worse performance of OR (+3.75%) when pre-trained with WOMD and fine tune on AV2
* There is a typo in line 196: two without.

**Robotics Focus:**

3

**Summary Of Paper:**

This paper presents JointMotion, a self-supervised pre-training method for joint motion prediction in self-driving vehicles. The main idea is to improve the prediction accuracy of traffic agents' future motion by learning joint scene representations and refining learned representations through complementary scene-level and instance-level objectives.

**Summary Of Recommendation:**

I recommend for a weak accept of this paper.

---

### Author Rebuttal · Authors · 2024-08-10

Thank you for your feedback on our work. In response to requests from reviewers, we have extended our experiments and updated our paper accordingly. In the new PDF version, changes are highlighted in yellow.

---

### Decision · Program_Chairs · 2024-09-04

**Decision:**

Accept

**Comment:**

### Strengths:

- Re: Methodology: R-ffHe and R-ZYgU mention innovation in the scene-level self-supervised learning (SSL) that connects motion and environment without requiring negative examples, enhancing scene-wide representation learning; they highlight effective application across different environment representations and information fusion mechanisms, and they highlight that the instance-level SSL reconstructs masked polylines for motion sequences, lanes, and traffic lights, refining detailed representations. R-ffHe mentions that the Adaptive Pre-Training Decoder enables generalization across various datasets and environment representations, showing adaptability in models with early fusion mechanisms.

- Re: Experiments: All reviewers cite the manuscript for having extensive experiments that demonstrate that JointMotion outperforms recent SSL methods and state-of-the-art (SOTA) methods in joint motion prediction. They highlight significant improvements over non-ensembling joint prediction and joint-prediction variants of marginal-prediction architectures. They suggest that the method demonstrates generalizability across different backbone models.

- Re: Clarity: R-ZYgU mentions that the manuscript is generally well-written, allowing readers to follow through the sections and understand the methodology and results.
- Re: Contribution: R-XhvA highlights the work for combining scene-level similarity learning and masked autoencoding, which they say suggests complementary properties and contributing to the field.


### Weaknesses:

- Re: Methodology:  R-ZYgU and R-XhvA express concerns over some methodological details that require familiarity with referenced models, suggesting a need for summarizing these details in supplementary materials. They also mention a lack of clarity in the pre-training procedures and the relationship between contributions and improvements.
- Re: Experiments: R-ZYgU and R-XhvA cite concerns over inconsistent experimental settings in Tables 2 and 3. R-XhvA expressed concerns over missing ablation studies for masked autoencoding from JointMotion, affecting the robustness of the results.
- Re: Clarity: R-ZYgU and R-XhvA cite concerns over the lack of clarity in the Figure 1 and Figure 3 descriptions; they mention that Figure 4’s training time comparison lacks clarity and might mislead readers. R-ffHe and R-XhvA assert that major revisions needed to improve organization and clarity and that specific sections, such as experiment setup and pre-training procedures, need better descriptions.
- Re: Novelty: R-ffHe and R-XhvA mention issues in originality, mentioning that scene-level similarity learning is not considered a major contribution due to extensive prior research and that combining existing techniques is seen as incremental rather than groundbreaking.

- Re: Limitations: All reviewers point out that the Limitations section is not well addressed, that the impact is considered incremental with no significant advances in robotics or machine learning, and that the reported improvements fall short in various SOTA metrics.

### Post-rebuttal Meta Review Statement

I applaud the reviewers and authors for engaging in a productive and efficient discussion that resulted in several improvements to the paper and a notable increase in a reviewer score.

While concerns still remain for this work (e.g., incremental contribution and lacking detailed analysis of convergence properties, discussed by R-XhvA), I do feel as if the stated strengths of the manuscript, in sum, sufficiently outweigh the limitations.

I recommend for Accept as Poster.

I encourage the authors to follow through with all the promised changes and to incorporate all additional reviewer requests, to improve the paper even further.